

# Ontogenetic progression of individual head size in the larvae of the beetle *Trypoxylus dichotomus* (Coleoptera: Scarabaeidae): catch-up growth within stages and per-stage growth rate changes across stages

Sugihiko Hoshizaki

Laboratory of Applied Entomology, Department of Agricultural and Environmental Biology, Graduate School of Agricultural and Life Sciences, The Univeristy of Tokyo, Yayoi, Bunkyo-ku, Tokyo, Japan

## ABSTRACT

The ontogenetic progression of insect larval head size has received much attention due to its fundamental and practical importance. However, although previous studies have analyzed the population mean head size, such an approach may not be appropriate for developmental studies of larval head sizes when the trajectory of individual head size growth is correlated with pre-molt head size and developmental stage. Additionally, there is covariation between the head and body sizes; however, few studies have compared the ontogenetic progression of individual head sizes with that of individual body sizes. In this investigation, the per-stage growth rates (PSGRs) for head width (HW) and cubic-rooted body mass at the beginning of each instar (body size, BS) were assessed in *Trypoxylus dichotomus*. Linear models were used to test the size- and instar-dependence of the ontogenetic progression of individual HW and BS. The individual PSGRs of the HW ($iPSGR_H$) and BS ($iPSGR_B$) were then compared. In addition, the allometric relationship between HW and BS was examined. The $iPSGR_H$ was negatively correlated with the pre-molt HW at every instar (*i.e.*, head catch-up growth). Furthermore, the mean $iPSGR_H$ at L2 was relatively higher than that at L1 when the pre-molt HW was used as covariate in the analysis (*i.e.*, instar-effect), whereas the mean $iPSGR_H$ decreased ontogenetically. The $iPSGR_B$ showed catch-up growth and instar-effects similar to those of $iPSGR_H$; however, $iPSGR_H$ was found to be lower than $iPSGR_B$. Due to the differences between the PSGRs for the larval head and body, the larval head size showed negative ontogenetic allometry against body size.

Corresponding author
Sugihiko Hoshizaki,
ahossy@g.ecc.u-tokyo.ac.jp

## INTRODUCTION

Physiological studies on insects and other animals have largely focused on the developmental control of body and body part sizes (*Sehnal, 1985*; *Nijhout, 1994*; *Minelli & Fusco, 2013*; *Nijhout et al., 2013*; *Klingenberg, 2016*; *Shingleton & Frankino, 2018*). The larval development of insects and other arthropods consists of distinct developmental stages (or instars) that are separated by molting events. The larval body size, often measured as fresh mass or total length, increases continuously during the feeding period within the instar, while the size of the exoskeleton parts, such as the head capsule and legs, only increases during the molting period. Consequently, the sizes of exoskeletal parts increase in a stepwise manner during larval development. In applied entomology, the size of larval head capsules and other hard parts of the exoskeleton have been used as indicators of body size and instars (*e.g.*, *Shimoda, Kamiwada & Kiguchi, 1994*; *Higo, Sasaki & Amano, 2022*; *Tanaka, 2022*).

Owing to its fundamental and practical importance, as described above, the ontogenetic progression of larval head size in insects has received much attention. Many previous studies have compared the per-stage growth rates (PSGRs) of the population mean larval head size (mPSGR$_H$) across instars. It is generally accepted that the mPSGR$_H$ is approximately constant across instars in a constant environment (Dyar's Rule) (*Dyar, 1890*; *Nijhout, 1994*; *Hutchinson et al., 1997*; *Daimon et al., 2021*). However, previous studies have also reported violations of Dyar's Rule in various insects, and in particular, there is a tendency for ontogenetic decreases in mPSGR$_H$ (*Clark & Hersh, 1939*; *Beck, 1950*; *Rodriguez & Maldonado, 1974*; *Hansen, Owens & Huddleson, 1981*; *Savopoulou-Soultani & Tzanakakis, 1990*; *Klingenberg & Zimmermann, 1992*; *Morales-Ramos et al., 2015*; *Hoshizaki, 2020*; *Springolo, Rigato & Fusco, 2021*; *Baraldi, Rigato & Fusco, 2023*).

However, analyses of mPSGR$_H$ alone may not be sufficient to understand the ontogenetic progression of individual head sizes. For mPSGR$_H$ to suffice in this regard, the PSGR of the head size of individual larvae (iPSGR$_H$, Fig. S1) must be invariant among conspecifics at the same instar, irrespective of their pre-molt head size (Figs. 1A and 1C). In contrast, iPSGR$_H$ negatively correlates with the pre-molt head size among conspecifics at the same developmental stage (head catch-up growth) in a few arthropods (*Tanaka, 1981*; *Fusco et al., 2004*; *Thompson, 2019*; *Fusco, Rigato & Springolo, 2021*). Combining the presence/absence of an ontogenetic decrease in mPSGR$_H$ and that of head catch-up growth could allow for the illustration of four patterns of the relationship between the pre-molt head width (HW) and iPSGR$_H$ (Fig. 1). Whether or not mPSGR$_H$ strictly follows Dyar's Rule, head catch-up growth could occur (Figs. 1B and 1D). Moreover, when mPSGR$_H$ decreases ontogenetically and head catch-up growth occurs, the iPSGR$_H$ to pre-molt HW relationships at a given stage and its subsequent stage may be constant or different (Fig. 1D). Thus, not only the conformity to Dyar's Rule in terms of mPSGR$_H$, but also the relationship between iPSGR$_H$ and HW, should be examined to understand the ontogenetic progression of larval head size.

Furthermore, catch-up growth within the instar and ontogenetic changes in mPSGR have also been found in the body size growth of insects (for body catch-up growth
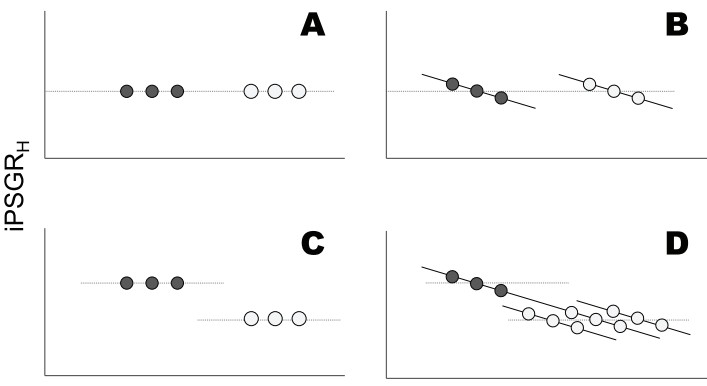

**Figure 1** **Relationships between the iPSGR$_H$ and pre-molt HW in three hypothetical individuals.** Dark and light gray represent a given molt and the subsequent molt, respectively. The dotted line indicates mPSGR$_H$. The mPSGR$_H$ is constant across the two molts in the top row (A and B), whereas it decreases ontogenetically in the bottom row (C and D). The iPSGR$_H$ for a molt is constant irrespective of the pre-molt HW in the left column (A and C), whereas it is negatively dependent on the pre-molt HW in the right column (B and D). In (D), the iPSGR$_H$-HW relationship at the later molt was illustrated in three ways, *i.e.*, the relationships higher than, identical to, and lower than the relationship at the earlier molt.

*Klingenberg, 1996*; *Hoshizaki, 2019*; for ontogenetic changes in body PSGR *Klingenberg & Zimmermann, 1992*; *Elmes et al., 2001*; *Grunert et al., 2015*; *Hoshizaki, 2020*; *Kivelä et al., 2020*). When head catch-up growth and/or ontogenetic changes in mPSGR$_H$ occur, head PSGR may or may not be equal to body PSGR. This can be determined by comparing the iPSGR$_H$ and the individual per-stage growth rate of the linear body size (iPSGR$_B$, Fig. S1). While a previous study has shown that the mPSGR$_H$ tends to be lower than the PSGR for the mean body size (mPSGR$_B$) in water striders (*Klingenberg & Zimmermann, 1992*), there are no studies comparing iPSGR$_B$ and iPSGR$_H$ to my knowledge.

Additionally, when a larval body part size and its body size grow differentially during a developmental stage, their scaling relationships would be affected at the subsequent stage. The scaling relationship between body part size and body size is referred to as allometry (*Huxley, 1932*; *Nijhout et al., 2013*; *Pélabon et al., 2013*; *Klingenberg, 2016*; *Shingleton & Frankino, 2018*). In particular, the scaling relationships concerning size variation between different developmental stages and size variation between conspecific animals at the same developmental stage are referred to as ontogenetic allometry and static allometry, respectively (Fig. S1). When larval iPSGR$_H$ and iPSGR$_B$ are not equal for a given instar, both the ontogenetic and static allometry of head size to body size likely deviates from the isometry at the beginning of the subsequent instar. Negative ontogenetic allometry of larval head size with respect to body size has been found in several arthropods, most of which are hemimetabolous insects (*Clark & Hersh, 1939*; *Brown & Davies, 1972*; *Rodriguez & Maldonado, 1974*; *Klingenberg & Zimmermann, 1992*; *Avendaño & Sarmiento, 2011*). In this article, I show negative ontogenetic allometry of larval head against body size in a holometabolous insect.

The Japanese rhinoceros beetle, *Trypoxylus dichotomus* (Coleoptera: Scarabaeidae), was used as the study system. I aimed at examining head catch-up growth and ontogenetic changes in head size growth, to assess whether such patterns for head size growth involve exoskeleton-specific mechanisms, and to examine the ontogenetic and static allometry of head against body sizes at the beginning of instars.

# MATERIALS AND METHODS

## Insects

*T. dichotomus* inhabits forests and farmlands on the main Japanese islands and has a univoltine life cycle in the field (*Araya et al., 2012*; *Kojima et al., 2020*). The larvae feed on decaying organic matter and pass through three instars before molting into pupae. Both the $mPSGR_H$ and $mPSGR_B$ decrease with the ontogeny (*Hoshizaki, 2020*). The body size of neonates varies among individuals with the age and size of their mothers (*Kojima, 2015*). Small neonates subsequently gain relatively more body mass than larger ones during every instar, *i.e.*, negative size-dependence occurs in $iPSGR_B$ (*Hoshizaki, 2019*).

Five pairs of *T. dichotomus* beetles were collected from Tokyo, Japan, in the summer of 2012. They were mated in the laboratory. The females were allowed to lay eggs, and the eggs were kept on pieces of moist cotton wool in Petri dishes until they hatched. The larvae were fed with hummus (Kokusankabutomatto, Dorcus Owners Shop, Osaka, Japan). The insects were reared at $25 \pm 1$ °C and with a light:dark ratio of 15:9. The three larval instars were denoted as L1, L2, and L3, and the pupal stage was denoted as P. The larvae were kept individually in plastic cups with volumes of 60, 430, and 860 mL during L1, L2, and L3, respectively. The sex of the insects was identified at either the late L3 or P stage. The details of rearing are described in a previous article (*Hoshizaki, 2019*).

## Morphological measurements

My study was based on a longitudinal dataset of the morphometric measurements of 51 male and 47 female insects. The HW was measured to 0.05 mm with calipers at every instar. Fresh body mass was measured with an electronic balance (PFB200-3; KERN & SOHN Gmbh, Balingen, Germany) after hatching before consuming food and immediately before or after ecdysis to L2 and L3. The final body mass at L3 was measured as previously described (*Hoshizaki, 2019*). The cubic root of the body mass measurement was used as the linear body size (BS) at the beginning of L1, L2, and L3 and at the end of L3. These data were previously used (*Hoshizaki, 2019*, *2020*) but in different ways and with objectives different from those in the present study.

## Statistical analyses

Statistical analyses were conducted using R version 3.6.2 (*R Core Team, 2019*). Data on the male and female insects were analyzed separately.

The individual per-stage growth rate of HW ($iPSGR_H$) was defined as the post-molt/pre-molt size ratio of individual larvae. The relationship between $iPSGR_H$ and pre-molt HW was examined using linear models in which log-transformed $iPSGR_H$ was set as the response variable. The log-transformed pre-molt HW and instar (L1 or L2) were set as

predictor variables. First, the model, including the predictor variables and their interactions, was analyzed. If these interactions were not significant, the model was analyzed excluding the interactions.

The variance of ln HW was calculated for each instar. The data were resampled 10,000 times using the bootstrap method, and the 95% confidence interval (CI) of the variance was estimated. The Brown–Forsythe test was conducted against the ontogenetic homogeneity of the variance (implemented in R package 'car').

Catch-up growth can be detected as a departure from the expected increase in size variance across ontogeny according to *Fusco, Rigato & Springolo (2021)*. The expected progression of ln HW variance in the absence of catch-up growth was estimated, as follows. For a log-transformed HW (X), the values of within-stage variance in two successive stages (i, i + 1) are bound by the following relationship

$$var(X_{i+1}) = var(X_i) + var(\rho_i) + 2 \times cov(X_i, \rho_i)$$

where $\rho_i$ is $X_{i+1} - X_i$. Assuming non-null var($\rho$) at each stage, var(X) tends to increase stage by stage in proportion to var($\rho$), unless this is compensated for by a negative cov(X, $\rho$). The expected progression of ln HW variance at L2 and L3 in the absence of catch-up growth was calculated by setting the expected size variance in L1 and L2, respectively, as equal to observed values, *i.e.*, var($X_1$) and var($X_2$), respectively, and adding the observed values of var($\rho_1$) and var($\rho_2$), respectively. This is equivalent to setting cov(ln HW, ln iPSGR$_H$) = 0 while maintaining the observed growth parameters. The 95% CI of the expected variance was estimated by bootstrap resampling (10,000 times).

The individual per-stage growth rate of body size (iPSGR$_B$) for instar N was defined as the BS at instar N + 1 divided by that at instar N. Note that the iPSGR$_B$ for L3 was calculated as the BS at the end of L3 divided by the BS at the beginning of L3. The relationship between iPSGR$_B$ and BS was examined using linear models. The log-transformed iPSGR$_B$ was set as the response variable. The log-transformed BS and the instar (*i.e.*, L1, L2, or L3) were set as predictor variables. First, the model, including predictor variables and their interactions, was analyzed. If the interactions were not significant, the model was re-analyzed excluding the interactions.

The iPSGR$_H$ and iPSGR$_B$ at the same developmental stage were compared using a paired *t*-test.

The allometry of HW against BS is expressed as a power-law function, *i.e.*, HW = $a \times BS^b$, where $a$ and $b$ are the parameters of the scaling factor and scaling exponents, respectively. If $b$ = 1, <1, or >1, the relationship is referred to as isometry, negative allometry, or positive allometry, respectively. Ontogenetic allometry for an individual at instar N, where N was I or II, was defined as the segment between two data points representing the beginning of instars N and N + 1 on the log-log plot of HW against BS (Fig. S1). The slope for instar N was calculated as (log HW$_{N+1}$ − log HW$_N$)/(log BS$_{N+1}$ − log BS$_N$), where HW$_{N+1}$ and HW$_N$ represented HW at instars N + 1 and N, respectively, and BS$_{N+1}$ and BS$_N$ represented body sizes at the beginning of instars N + 1 and N, respectively. The slope was tested against the null hypothesis of 1 using the *t*-test for the different instars. The slopes of L1 and L2 were compared using a paired *t*-test. Static allometry was calculated using linear models at
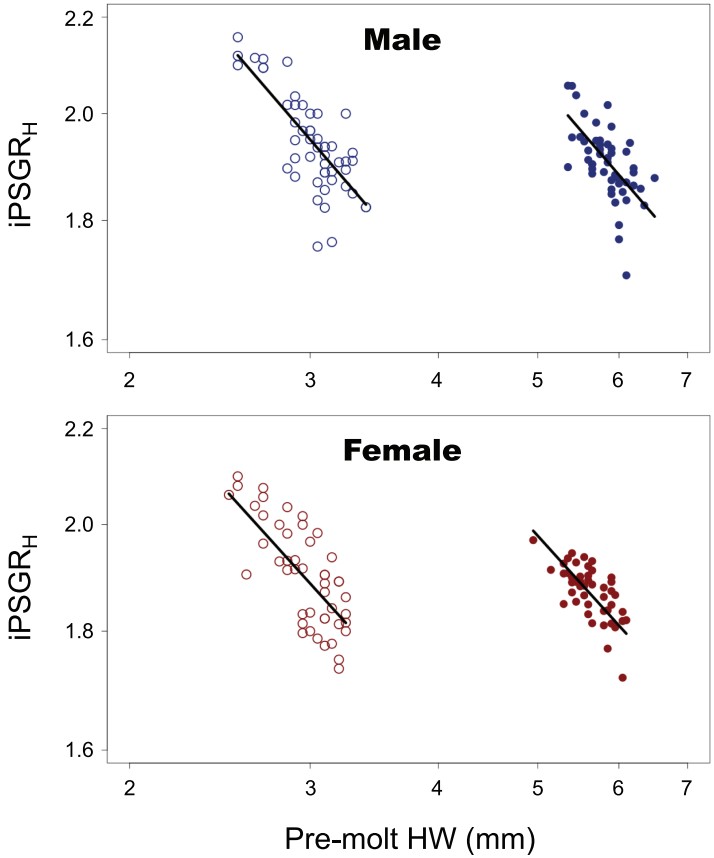

**Figure 2 The relationship between the iPSGR_H and the pre-molt HW in *T. dichotomus*.** Open and closed circles represent L1 and L2, respectively. Solid lines indicate the fitted linear models. The X- and Y-axes are indicated in the log scale.

the beginning of each instar stage (Fig. S1). Log-transformed HW was set as the response variable. The log-transformed BS was used as a predictor variable. The slope was tested against the null hypothesis of 1 using a *t*-test.

## RESULTS

For both male and female insects, the larger the pre-molt HW, the smaller the iPSGR_H at every instar (Fig. 2). The interaction of the predictor variables in the full model was not significant for either sex (male insects: $t = 0.516$, $p = 0.61$; female insects: $t = 1.51$, $p = 0.13$). When the models without interactions were analyzed, the slope was significantly negative for both sexes (male insects: $t = -10.4$; $p < 0.0001$; female insects: $t = -10.4$, $p < 0.0001$). The effect of instar was significant in both sexes (male insects: $t = 9.59$, $p < 0.0001$ female insects: $t = 9.61$, $p < 0.0001$); the mean iPSGR_H for L2 was larger than that for L1 when the pre-molt head size was taken into account (Fig. 2).

The variance of log-transformed HW significantly decreased along the instar progression in both sexes (Fig. 3; male insects: Brown–Forsythe test, $W = 6.98$, $p = 0.001$; female insects: Brown-Forsythe test, $W = 10.3$, $p < 0.0001$). The 95% CIs for L1 and L2 did not overlap with each other, while those for L2 and L3 did overlap slightly. The expected
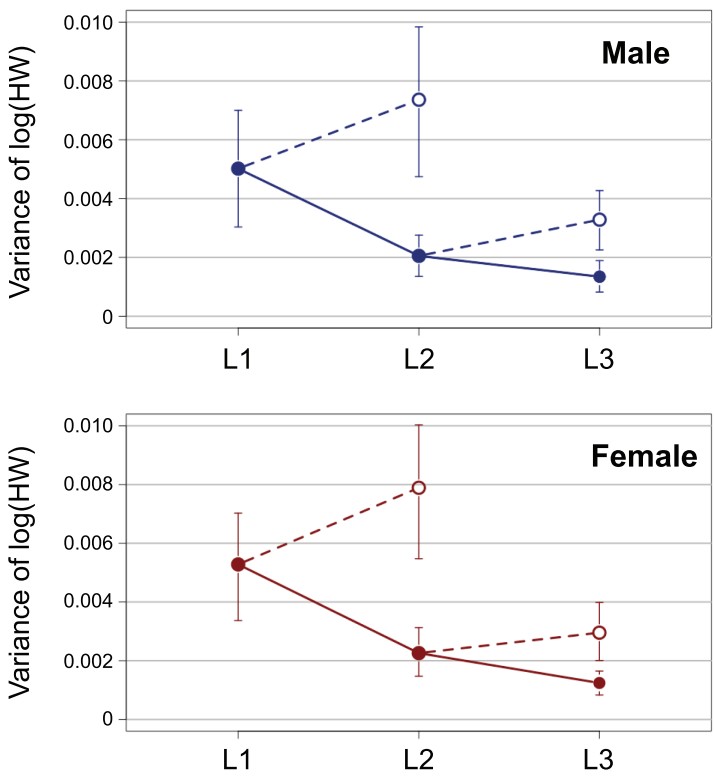

**Figure 3 Observed ontogenetic progression of head size variance (solid circles) in *T. dichotomus* larvae, compared with the expected progression of head size variance in the absence of catch-up growth (open circles).** Vertical bars indicate 95% confidence intervals.

variance of log-transformed HW in the absence of catch-up growth was higher than the observed value for both L2 and L3 in both sexes; the 95% CIs for the former and the latter did not overlap with each other at both L2 and L3 (Fig. 3).

In the analyses of iPSGR$_B$, the interaction of the predictor variables in the full model was not significant for either sex (BS for male insect $^*$ L1: $t = -1.35$, $p = 0.18$; BS for male insect $^*$ L3: $t = -0.269$, $p = 0.79$; BS for female insect $^*$ L1: $t = -0.836$, $p = 0.41$; female for BS $^*$ L3: $t = -0.466$, $p = 0.64$). When the models without interactions were analyzed, the effect of instar was significant in both sexes (L1 for male insects: $t = -4.47$, $p = 1.5$ E−3; L3 for male insects: $t = 7.47$, $p < 0.0001$; L1 for female insects: $t = -3.55$, $p = 0.0005$; L3 for female insects: $t = 6.15$, $p < 0.0001$). Namely, the mean of iPSGR$_B$ at L2 and L3 was significantly higher than that at L1 and L2, respectively, in both sexes (Fig. 4).

iPSGR$_H$ was found to be significantly lower than iPSGR$_B$ at both L1 and L2 (Fig. 5; paired $t$-test for male insects at L1: $t = -29.8$, $p < 0.0001$; paired $t$-test for female insects at L1: $t = -21.9$, $p < 0.0001$; paired $t$-test for male insects at L2: $t = -17.1$, $p < 0.0001$; paired $t$-test for female insects at L2: $t = -12.8$, $p < 0.0001$).

The slope of ontogenetic allometry was significantly less than 1 in both L1 and L2 (Fig. 6; $t$-test for L1 in female insects, $t = -21.5$, $p < 0.0001$; $t$-test for L1 in male insects, $t = -29.4$, $p < 0.0001$; $t$-test for L2 in female insects, $t = -12.6$, $p < 0.0001$; $t$-test for L2 in male insects, $t = -17.0$, $p < 0.0001$), and for both sexes, the slope in L1 was found to be

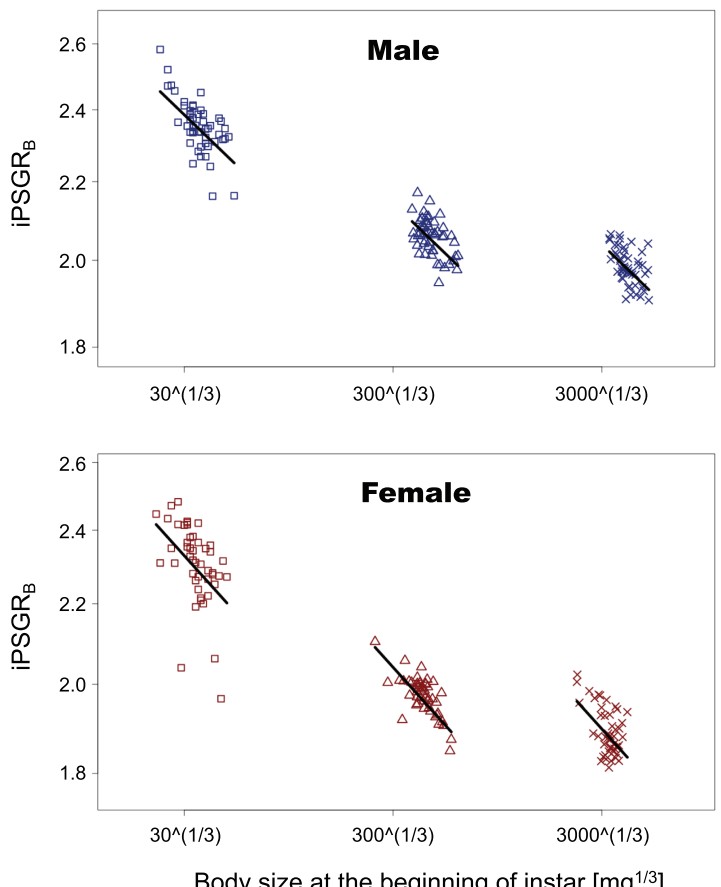

**Figure 4 The relationship between iPSGR$_B$ and body size at the beginning of instars in *T. dichotomus*.** Squares, triangles, and crosses represent L1, L2, and L3, respectively. Solid lines indicate the fitted linear models. The X- and Y-axes are presented in the log scale.

significantly smaller than that in L2 (paired *t*-test for female insects, $t = -11.7$, $p < 0.0001$; paired *t*-test for male insects, $t = -10.5$, $p < 0.0001$).

The slope of the static allometry was observed to be slightly less than 1 at the beginning of every instar (Table 1). It was of note, however, that the slope was significantly less than 1 only for male insects at L3 and female insects at L2 and L3.

## DISCUSSION

The relationship between iPSGR$_H$ and pre-molt HW in *T. dichotomus* larvae was examined. The iPSGR$_H$ was negatively correlated to the pre-molt HW at both L1 and L2. The variance of head size decreased ontogenetically and was smaller than the expected variance in the absence of head catch-up growth. This indicates catch-up growth in the head within L1 and L2. Simultaneously, the effect of instar on the iPSGR$_H$ was significant; the mean iPSGR$_H$ at L2 was significantly larger than that at L1 as for HW. Similarly, the iPSGR$_B$ was negatively correlated to the body size at the beginning of each instar, and the mean iPSGR$_B$ at an instar was significantly larger than that at the previous instar when the body size at the beginning of the instar was taken into account.

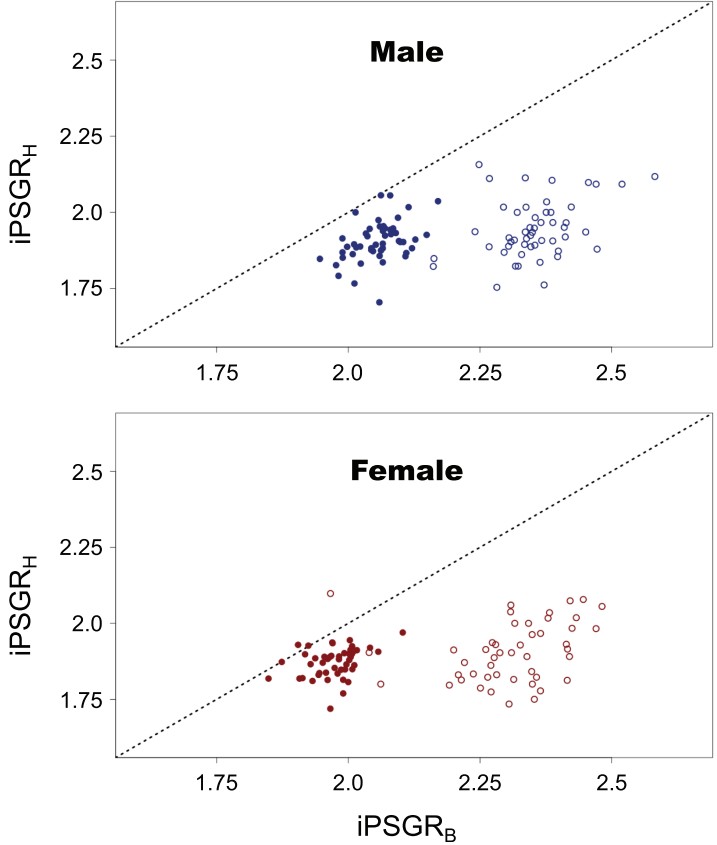

**Figure 5 The iPSGR$_H$ and iPSGR$_B$ in *T. dichotomus*.** Open and closed circles represent L1 and L2, respectively. When the iPSGR$_H$ and iPSGR$_B$ for an individual are equal, the point is positioned on the dotted line.

Thus, iPSGR$_H$ and iPSGR$_B$ in *T. dichotomus* showed similar patterns, *i.e.*, size-correlation and the instar-effect (ontogenetic effect). This is not very surprising as it is sure that insect larval head sizes and body mass covary. In general, insect larval head growth occurs after body mass growth within instar is completed. This implies that the degree of body mass increase within instar would affect that of head size increase at the subsequent molting in insects; however, the degree of head size increase is not equal to that of body mass increase in *T. dichotomus* as the present study found that iPSGR$_H$ was smaller than iPSGR$_B$.

Empirical support for Dyar's Rule has relied on the mean head size growth. However, conformities to Dyar's Rule based on the mean head growth do not preclude the occurrence of size-correlation and/or stage-effects in head size growth at the individual level (Fig. 1). In *T. dichotomus*, while the mPSGR$_H$ approximately conforms to Dyar's Rule (*Hoshizaki, 2020*), both size-correlation and the stage-effect occurred in iPSGR$_H$. This finding corresponds to Fig. 1D. It remains to ascertain how much such cases are common among other insects.

Previous studies have also reported deviations from Dyar's Rule in insects in terms of the mPSGR$_H$ and mPSGR$_B$; in particular, there has been a general tendency for an

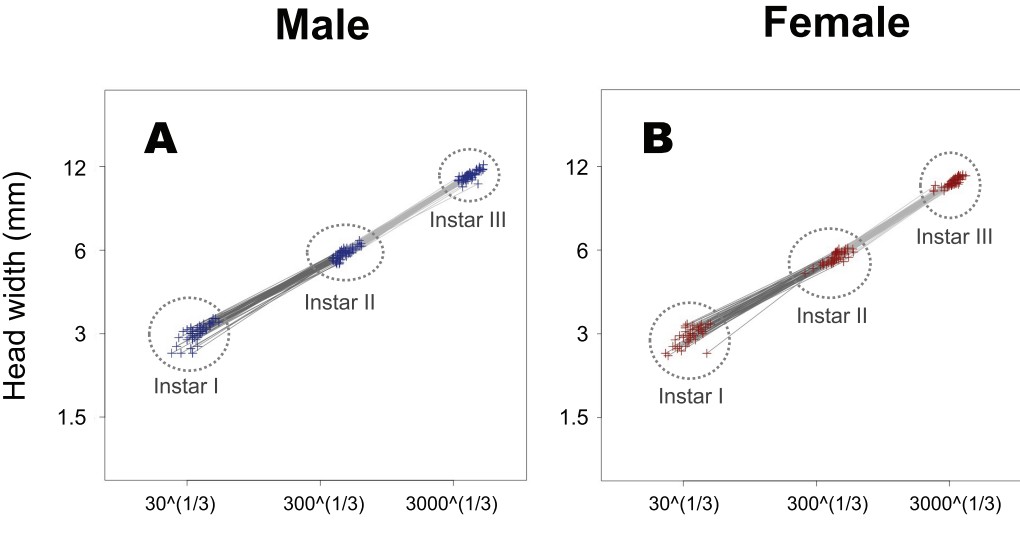

Body size at the beginning of instars (mg$^{1/3}$)

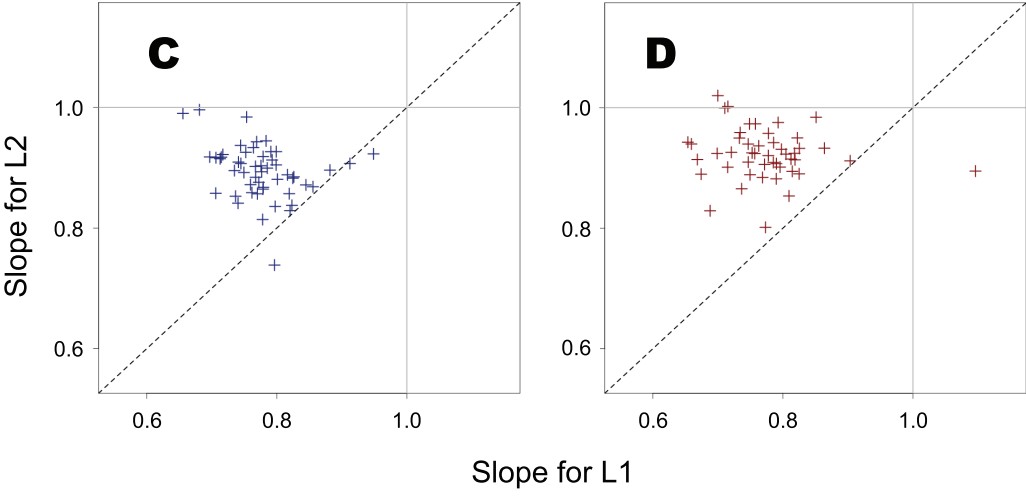

**Figure 6 Ontogenetic allometry of HW against body size at the beginning of instars in** ***T. dichotomus.*** (A and B) Individual ontogenetic allometry. Dark and light grey segments represent the ontogenetic allometry for L1 and L2, respectively. The X- and Y-axes are presented in the log scale. (C and D) Comparison of the allometric slopes for L1 and L2. When the two slopes for an individual are equal, the point can be seen on the dotted line.           

ontogenetic decrease in larval mPSGR$_H$ (mPSGR$_H$, *Savopoulou-Soultani & Tzanakakis, 1990*; *Klingenberg & Zimmermann, 1992*; *Morales-Ramos et al., 2015*; *Springolo, Rigato & Fusco, 2021* and references therein; mPSGR$_B$, *Klingenberg & Zimmermann, 1992*; *Elmes et al., 2001*; *Grunert et al., 2015*; *Kivelä et al., 2020*). Dyar's Rule can be used as a base of comparison against which specific adaptive hypotheses can be tested (*Klingenberg & Zimmermann, 1992*); however, in such studies, relying only on the mean size of the head (and also the body) can lead to inadequate hypotheses, as shown in the present study. It is important to examine not only mean but also individual sizes for a better understanding of

**Table 1 Static allometry slope at the beginning of each instar tested against isometry.**

| Sex | Instar | Slope | $t$-value | DF | $p$-value |
|---|---|---|---|---|---|
| Male | L1 | 0.92 | 0.72 | 49 | 0.477 |
| | L2 | 0.84 | 1.92 | 49 | 0.060 |
| | L3 | 0.72 | 3.17 | 49 | 0.003 |
| Female | L1 | 0.76 | 1.69 | 45 | 0.098 |
| | L2 | 0.73 | 3.77 | 45 | <0.001 |
| | L3 | 0.72 | 4.03 | 45 | <0.001 |

the mechanisms underlying the growth of the head capsule, other exoskeleton parts, and the whole body of insect larvae.

In general, the differential PSGR between a body part and the whole body during a developmental stage most likely affects the scaling relationship between the sizes of the part and whole at the subsequent stage. (i) The $iPSGR_H$ was lower than the $iPSGR_B$ at both L1 and L2 in *T. dichotomus*. Consequently, the relative size of the head is decreases ontogenetically at the beginning of the instar, as shown by another finding that the slopes of individual ontogenetic allometry were less than 1 at both L1 and L2. (ii) In addition, the difference between $iPSGR_H$ and $iPSGR_B$ at L2 decreased compared to that at L1. This resulted in that single larvae at L3 have relatively larger heads than at L2, as shown by the result that the individual ontogenetic allometry slope at L2 was larger than that at L1. (iii) The $iPSGR_H$-HW relationship had a slope of less than 1 for both L1 and L2. Considering that conspecific larvae with small heads most likely have small bodies, larger-bodied individuals may have relatively smaller heads than smaller-bodied ones at the same developmental stage. This is consistent with the finding that the slope of the HW-BM static allometry was significantly less than 1 at the beginning of L3 in male and female insects and L2 in female insects, and also tended to be so at the beginning of the other combinations of sex and instar. The results of this study are consistent with the mechanism by which the differential PSGR of head and body size affects head-to-body allometry in insect larvae. A previous study has shown that the larval head shape of the holometabolous insect *Pieris brassicae* changes ontogenetically (*Springolo, Rigato & Fusco, 2021*). Allometric relationships between larval head and body sizes in holometabolous insects have received little attention.

The old exoskeleton in insect larval molts serves as the template for the new exoskeleton (*Bennet-Clark, 1971*). However, the per-stage growth rate of the head capsule is not constant but negatively size-dependent in at least a few insects (*Tanaka, 1981*; *Fusco et al., 2004*; *Fusco, Rigato & Springolo, 2021*; the present study). *T. dichotomus* larvae with smaller heads and bodies at the beginning of the instar (relative to those with larger heads and bodies) tended to accumulate relatively more mass (*i.e.*, developmental resources) in their bodies during the feeding period in that instar (*Hoshizaki, 2019*; Fig. 4 in the present study). They then became equipped with relatively larger heads at the beginning of the subsequent instar as shown by <1 slopes of ontogenetic allometry of HW against BS (Fig. 6

in the present study). In general, new head capsules of larval insects form under the old ones after body mass growth finishes (*Nijhout, 1994*). Taken together, this suggests that *T. dichotomus* larvae with smaller heads and bodies at the beginning of young instars (relative to those with larger heads and bodies) allocate a higher proportion of their developmental resources to the head exoskeleton at the subsequent larval molt, *i.e.*, a hypothesis that the degree of developmental resource accumulation during the feeding period of instar influences iPSGR$_H$ at the subsequent molt. Another hypothesis would be possible; the higher iPSGR$_H$ in *T. dichotomus* larvae with smaller heads and bodies at the beginning of young instars could be caused by increasing the duration of that instar. In the butterfly *P. brassicae*, catch-up growth in the larval head size was not attained through the regulation of instar duration (*Fusco, Rigato & Springolo, 2021*). It may be relevant for future studies to consider these hypotheses when studying the mechanisms underlying size-correlation and the instar-effect of larval head size growth in insects.

## CONCLUSIONS

In conclusion, (1) the individual head size shows catch-up growth in L1 and L2 in *T. dichotomus* larvae; (2) the mean iPSGR$_H$ at L2 was relatively higher than that at L1 when the pre-molt HW was used as covariate in the analysis (instar-effect), whereas the mean iPSGR$_H$ decreased ontogenetically; (3) similar catch-up growth and instar-effect occur in the relationship of iPSGR$_B$ to body size; (4) iPSGR$_H$ was less than iPSGR$_B$ at both L1 and L2; and (5) as a consequence of the differential PSGR of head and body within instars, the larval head-to-body ontogenetic and static allometries are/tend to be negative. Future work should examine whether size- and stage-dependence in larval head and body growth trajectories occur in insects other than *T. dichotomus*.

Finally, as a limitation of the present study, genetic and environmental components could not be distinguished in the catch-up growth in the head and body growth and head-to-body allometries in the *T. dichotomus* larvae. The patterns of catch-up growth possibly differ depending on the cause of size-variation at the beginning of a particular instar; larvae that are smaller-than-average for a genetic reason may grow differently from those that are smaller-than-average for an environmental reason. While the body size of neonates varies among individuals with the age and size of their mothers (*Kojima, 2015*), genetic variation in neonate size has not been found in *T. dichotomous*. Future work should address these subjects.

## ACKNOWLEDGEMENTS

Yukio Ishikawa, Takashi Matsuo, and Masami Shimoda helped maintain the facilities. Three anonymous referees provided constructive comments on previous versions of the article. Taiju Hoshizaki and Nobuko Hoshizaki allowed the author to spend weekends and holidays in the laboratory for this study. I would like to thank Editage for English language editing.

### Funding

The author received no funding for this work.

### Competing Interests

The author declares that they have no competing interests.

### Author Contributions

- Sugihiko Hoshizaki conceived and designed the experiments, performed the experiments, analyzed the data, prepared figures and/or tables, authored or reviewed drafts of the article, and approved the final draft.

### Data Availability

The raw measurements are available in the Supplemental Files.

### Supplemental Information

Supplemental information for this article can be found online at http://dx.doi.org/10.7717/peerj.15451#supplemental-information.

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
