# Peer review of "Ontogenetic progression of individual head size in the larvae of the beetle Trypoxylus dichotomus (Coleoptera: Scarabaeidae): catch-up growth within stages and per-stage growth rate changes across stages"

_PeerJ, doi:10.7717/peerj.15451_

## Round 0.1 · original submission · Major Revisions

Dear Dr. Hoshizaki:

Thanks for submitting your manuscript to PeerJ. I have now received three independent reviews of your work, and as you will see, one reviewer (R3) recommended rejection, while the other two suggested minor revision (yet with many suggested changes). I am affording you the option of revising your manuscript according to all three reviews.

The greatest concern is the treatment of growth; this should be thoroughly addressed in your revision. Please be more thorough with description of your analyses. Ensure that all of the data is included and easy to obtain. Your work should be repeatable. A more inclusive background should be provided in the Introduction, with relevant literature cited.

The reviewers raised many other concerns about the manuscript. Please address all of these issues.

I look forward to seeing your revision, and thanks again for submitting your work to PeerJ.

Good luck with your revision,

-joe

Reviewer 1 ·

Basic reporting

Perfect technically.

Experimental design

No problems.

Validity of the findings

All OK, some problems with interpretation, see below.

Additional comments

This is an extremely careful paper which reaches rather straightforward and non-surprising but nevertheless useful conclusions. All seems to be OK technically. I do not have much criticism to offer, just perhaps the following:

+ smaller larvae grow more, i.e. they show compensatory or catch-up growth, OK. However, this paper completely ignores the question WHY were the small larvae small, and whether the parameters of the compensatory growth depend on the reason of being small. More specifically, if the reason was genetic then your results are interpretable as genetic correlations between initial and final size of an instar, if the reasons were environmental, then we saw environmental correlations which can, in turn, be subdivided according to the environmental factor (e.g. food quality or temperature) which was affecting larval size. In general, there is no reason to assume that genetic correlations and environmental correlations between two traits should be identical, or even similar to each other. I perfectly understand that you cannot separate the genetic and environmental components in your data but I suggest that you explicitly consider this limitation of your work in the Discussion, and tell that future studies should address this.

+ I am not really sure that you can interpret your data so that there are regulation mechanisms specific to head size if you compare the allometries of head and whole body only. This leaves the possibility open that all chitinized parts of the body are still regulated in the same way but it is just different for soft parts of the larval body. As the soft parts largely consist of reserves (including water) it appears likely that exactly the allometry of the soft parts is a different story compared to the allometry of the sclerotized parts. Please adjust your conclusions accordingly.

+ may you use ’growth ratio’ instead of ’growth rate’? ’Growth rate’ is frequently understood as including the dimension of time (e.g. mg per day) which appears not to be the case in this study.

Reviewer 2 ·

Basic reporting

The article is clear and well written. The only references I would suggest adding are the following:

Thompson, D. B. (2019). Diet-Induced Plasticity of Linear Static Allometry Is Not So Simple for Grasshoppers: Genotype–Environment Interaction in Ontogeny Is Masked by Convergent Growth. Integrative and Comparative Biology, 59(5), 1382-1398.

The figures are clear and readable and the results are insightful.

Experimental design

The experimental design is simple to follow and the methods of analysis are sufficient.

Validity of the findings

The findings are very important for the study of allometry and are clear, concise, and impart an important message to the community.

Reviewer 3 ·

Basic reporting

See attached file

Experimental design

See attached file

Validity of the findings

See attached file

Additional comments

See attached file

Annotated reviews are not available for download in order to protect the identity of reviewers who chose to remain anonymous.

---

## Round 0.2 · Minor Revisions

Dear Dr. Hoshizaki:

Thanks for revising your manuscript. The reviewers are mostly satisfied with your revision (as am I). Great! However, there are a few issues still to entertain. Please address these ASAP so we may move towards acceptance of your work.

Reviewer 3 kindly provided feedback on a marked-up version of the manuscript.

Best,

-joe

Reviewer 1 ·

Basic reporting

A careful work throughout, and my previous comments have been addressed. Nevertheless, a few points to add:

Line 25. This sentence “Additionally, body growth has not been widely considered in previous head growth studies, despite there being covariation between the head and body sizes.” is not too clear. Obviously there is a correlation between head and body sizes, but how does this fact relate to the first part of the sentence?

The paragraph starting at line 154 „According to Furco….“. Please add a couple of sentences telling us why was all this done. The relevence of analysing the variances in the present context is not immediately obvious and an introduction is needed.

The final paragraph, starting at line 326. Thank you for adding this but I would still add a few sentences to make the point more clear, perhaps using simpler wording. In particular, my main point would be that the patterns of compensatory (catch-up) growth may well differ depending on the reason of smaller-than-average size at a particular developmental stage. The larvae which are small for genetical reasons may grow differently from larvae which are small because of nutritional stress for example. In other words, compensation may depend on the factor which has caused the need to compensate.

Experimental design

OK, with the exception that genetic end environmental effects cannot be separated. This is properly acknowledged now.

Validity of the findings

All OK.

Additional comments

no comment

Reviewer 3 ·

Basic reporting

See attached file

Experimental design

This is a revised submission. I have already commented on that.

Validity of the findings

This is a revised submission. I have already commented on that.

Annotated reviews are not available for download in order to protect the identity of reviewers who chose to remain anonymous.

---

## Round 0.3 · accepted · Accept

Dear Dr. Hoshizaki:

Thanks for revising your manuscript based on the concerns raised by the reviewers. I now believe that your manuscript is suitable for publication. Congratulations! I look forward to seeing this work in print, and I anticipate it being an important resource for groups studying insect developmental biology. Thanks again for choosing PeerJ to publish such important work.

Best,

-joe